# Vegetation type is an important predictor of the arctic summer land surface energy budget

Despite the importance of high-latitude surface energy budgets (SEBs) for land-climate interactions in the rapidly changing Arctic, uncertainties in their prediction persist. Here, we harmonize SEB observations across a network of vegetated and glaciated sites at circumpolar scale (1994–2021). Our variance-partitioning analysis identifies vegetation type as an important predictor for SEB-components during Arctic summer (June-August), compared to other SEB-drivers including climate, latitude and permafrost characteristics. Differences among vegetation types can be of similar magnitude as between vegetation and glacier surfaces and are especially high for summer sensible and latent heat fluxes. The timing of SEB-flux summer-regimes (when daily mean values exceed 0 Wm$^{-2}$) relative to snow-free and -onset dates varies substantially depending on vegetation type, implying vegetation controls on snow-cover and SEB-flux seasonality. Our results indicate complex shifts in surface energy fluxes with land-cover transitions and a lengthening summer season, and highlight the potential for improving future Earth system models via a refined representation of Arctic vegetation types.

As the Arctic warms at a fast pace above the global average[1,2], changes to a broad range of biogeophysical variables are being reported and projected[3,4]. These changes include increases in rainfall, permafrost temperatures and permafrost thaw[4,5], declines in ice mass, snow thickness and -spatial extent[3,4], as well as complex changes in the composition, structure and function of Arctic vegetation, including shrub abundance, plant height and vegetation productivity[3,6–10].

These changes impact climate dynamics at local to global scales, through various land-atmosphere feedbacks[4,11] that are mediated by the land surface energy budget[12,13] (SEB). Simply, the SEB is defined with the most relevant SEB-components:

$$R_{net} = SW_{net} + LW_{net} = H + LE + G + M \qquad (1)$$

where $R_{net}$ is the net radiative energy absorbed (or lost) by the surface and $SW_{net}$ and $LW_{net}$ are surface net shortwave and longwave irradiances. H is the sensible heat flux, LE is the latent heat flux (excluding latent heat of fusion, which is separately denoted by "M") and G is the

ground heat flux through snow, ice or soil (all units in W m$^{-2}$, Supplementary Table 1)[13–15].

Despite their importance, uncertainties in SEB-components persist in high-latitude climate projections, specifically in the case of sensible and latent heat fluxes[16]. These turbulent fluxes can directly feed back to Arctic biophysical variables by affecting near-surface atmospheric temperature and humidity[14].

The magnitude and seasonality of SEB-components depend on a complex interplay of drivers, such as vegetation type, snow cover, soil and permafrost characteristics, topography, and meteorological conditions including cloud cover[11,13]. However, to date, a quantitative understanding of the importance of vegetation type compared to other drivers of the Arctic SEB is missing[11,15,17]. Land surface components in current Earth system models often represent Arctic vegetation by only a single or few plant functional types (PFTs)[18,19], despite the notable diversity thereof[20]. Previous observational studies demonstrate that Arctic vegetation types can influence SEB-components, including latent and sensible heat fluxes[11,15,21–27]. However, these studies

✉ e-mail: jacqueline.oehri@gmail.com; gabriela.schaepman@ieu.uzh.ch

either focus on qualitative descriptions of conditions at different sites, or only cover limited geographic extents.

Therefore, here, we provide a quantitative, circumpolar assessment of the observed SEB over treeless land >60°N in the time period 1994–2021 and compare the predictive skill of vegetation type with other important SEB-drivers related to climate, topography, snow cover, permafrost characteristics and cloud cover (see Supplementary Table 1 for a full list of SEB-drivers identified in this study through a systematic literature review). Specifically, we harmonize in situ observations from regional and global monitoring networks[28–36] (number of sites: 64, number of site-years: 652; Supplementary Tables 2, 3). Using literature-based site descriptions, we classify the vegetation type at each site according to the categories of the Circumpolar Arctic Vegetation map (CAVM[20]), the most thematically detailed classification currently available at the circumpolar scale[37]. These CAVM classes describe barren complexes (≤40% horizontal plant cover), graminoid tundra, prostrate dwarf-shrub tundra, erect-shrub tundra and wetland complexes[20]. Additionally, our data contained sites classified as glacier and boreal peat bogs, the latter of which occur south of the Arctic tree line but largely lack tree cover (Supplementary Fig. 1–3). For each site, we derived additional climatic and biophysical SEB-drivers using spatial data products (Supplementary Fig. 4). With these data, we quantify the magnitude and seasonality of surface energy fluxes and compare the relative importance of vegetation type and other SEB-drivers for explaining variations in SEB-patterns. We especially focus on the summer season (June–August), which is when local controls by the land surface are expected to be more important compared to the winter season[27,38].

## Results

### Importance of vegetation types for the terrestrial Arctic SEB

We use a variance partitioning analysis to compare the ability of 15 selected SEB-drivers (Supplementary Table 1) to predict summer mean magnitudes of the surface energy fluxes $R_{net}$, H, LE and G (Fig. 1; vegetation subset of data excluding glacier). This variance partitioning method allows us to account for the statistical confounding of predictors by discerning variance in surface energy fluxes explained jointly or independently by different drivers (see Methods).

We show that vegetation type is an important predictor of summer surface energy flux magnitudes. This importance is specifically pronounced for H and LE (Fig. 1b, c) for which vegetation type explains on average 56.3% (range: 53.8–58.8%) and 71.7% (range: 61.4–81.9%) of variance, respectively.

For G (Fig. 1d), vegetation type is ranked among the top three predictors (average explained variance 31.7%); after landscape-scale dominant vegetation ('CAVM type'; 40.6%) and bioclimatic subzone ('CAVM subzone'; 39.1%). These CAVM[20]-derived variables are based on satellite-observed reflectance (including the normalized difference vegetation index, NDVI[20]) and Arctic phytogeographic zones in combination with summer land surface temperatures, respectively (Methods).

For $R_{net}$ (Fig. 1a), latitude and snow cover duration are the most important predictors, whereas vegetation type shows intermediate importance. However, results from a supplementary analysis with normalized fluxes (i.e., daily mean fluxes divided by daily maximum potential incoming shortwave radiation; Methods) show that vegetation type again ranks among the three most important predictors for $R_{net}$ (Supplementary Fig. 5), together with bioclimatic subzone and landscape-scale dominant vegetation. The predictive ability of vegetation type for normalized fluxes of H, LE and G is similar to that of non-normalized fluxes (Supplementary Fig. 5). Vegetation type is among the top three predictors for normalized $SW_{net}$ (n.$SW_{net}$) and even the most important predictor for normalized $LW_{net}$ (n.$LW_{net}$). Vegetation type is also an important predictor of albedo and surface temperature ($T_{surf}$), which are directly related to $SW_{net}$ and $LW_{net}$[14] (Supplementary Fig. 5).

### Magnitude of the terrestrial Arctic SEB across land cover types

Using a linear mixed-model analysis (Methods), we estimate the mean (±95% confidence interval) surface energy fluxes for the terrestrial

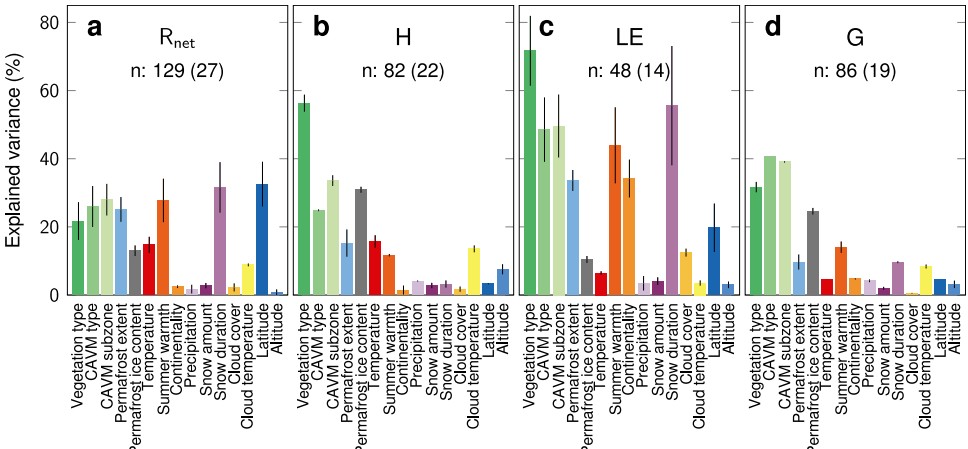

**Fig. 1 | Relative importance of 15 drivers of the surface energy budget (SEB) for average summer surface energy flux magnitudes at non-glacier sites.** Bars show the mean (bar height) and range (lines for each bar) of explained variance (%) averaged across all possible models with 2 predictors for each SEB-driver and corresponding summer magnitudes of surface energy fluxes (Wm⁻²): (**a**) $R_{net}$: net radiation, (**b**) H: sensible heat flux, (**c**) LE: latent heat flux, (**d**) G: ground heat flux. SEB-drivers: Vegetation type (dark green): local-scale, in situ vegetation type; CAVM type (green): landscape-scale, dominant vegetation type (surrounding area with radius of 500 m); CAVM subzone (light green): bioclimatic subzone; Permafrost extent (light blue): permafrost spatial extent; Permafrost ice content (grey): permafrost ground ice content; Temperature (red): mean annual air temperature; Summer warmth (dark orange): summer warmth index; Continentality (orange): Conrad's continentality index; Precipitation (light purple): mean annual precipitation; Snow amount (dark purple): mean annual snow water equivalent; Snow duration (purple): median annual snow cover duration; Cloud cover (light orange): mean cloud cover; Cloud temperature (yellow): mean cloud-top temperature; Latitude (dark blue): latitude (WGS84); Altitude (light blue): mean altitude (surrounding area with radius of 500 m; see Methods and Supplementary Table 1). n: average nr. of site years with average nr. of sites in parentheses. Results are based on the vegetation subset of our data (excluding glacier sites): nr. of sites: 31, nr. of site years: 234, period: 1994–2021. We repeated the analysis with additional surface energy fluxes, including normalized fluxes expressed as percentage of maximum potential incoming shortwave radiation (indicated with "n."-prefix in Supplementary Fig. 5). Source data are provided as a Source Data file.

Arctic (including vegetated and glacier sites) in summer (June–August) and across the entire year (Table 1). We find that vegetation type has a significant relationship with the mean magnitude of H, LE, $SW_{net}$ and $LW_{net}$ in summer ("Effect significance" in Table 1). At the yearly time-scale, we find a statistically significant relationship of vegetation type with $R_{net}$ (Table 1). For other surface energy fluxes including H and LE, available data is incomplete at this timescale, which hinders a proper assessment of their yearly magnitude and of the SEB closure (Supplementary Discussion).

Using a post-hoc analysis of the linear mixed-model results (Methods), we test the pairwise differences of summer surface energy flux magnitudes among vegetation types. We find strong differences for H, LE and $LW_{net}$ among vegetation types, which are of similar or even higher magnitude than differences between glacier and vegetation types (Supplementary Table 4). For example, for summer LE, the strongest difference (P-value < 0.001) is found between barren tundra (mean estimate = 0 W m$^{-2}$) and boreal peat bog (mean estimate = 75 W m$^{-2}$). For summer H, almost equally strong differences (P-value < 0.05) are found between barren tundra (mean estimate = −18 W m$^{-2}$) and erect-shrub (mean estimate = 38 W m$^{-2}$) as well as prostrate-shrub tundra (mean estimate = 39 W m$^{-2}$), respectively. For summer $LW_{net}$, we again find the strongest differences between barren tundra (mean estimate = −71 W m$^{-2}$) and erect-shrub tundra (mean estimate = −29 W m$^{-2}$; P-value < 0.01). Most significant pairwise differences among vegetation types include comparisons to either barren or boreal peat bog and differences among prostrate-shrub tundra, erect-shrub tundra and wetlands are not or only marginally significant (Table 1; Supplementary Table 4).

The ratio of H to LE (i.e., the Bowen ratio) in summer is >1 for prostrate-shrub tundra, erect-shrub tundra and wetlands (Bowen ratio = 1.6, 1.5 and 1.1, respectively), whereas it is <1 for graminoid tundra, boreal peat bog and barren tundra (Bowen ratio = 0.6, 0.1 and <0, respectively; Bowen ratios calculated from summer H and LE values of Table 1).

In a supplementary analysis with normalized fluxes and the vegetation data subset only (excluding glaciers, Supplementary Table 5), we find that vegetation type effects on H and LE are significant for each summer month (June–August). For other surface energy fluxes, we find that effect significance changes throughout the summer season; for example, vegetation type is significantly related to albedo and $R_{net}$ in June and July, respectively, and to G in August (Supplementary Table 5).

## Seasonality of the Arctic SEB across vegetation types

The cumulative summer energy budget depends on both the magnitude and the seasonality of SEB fluxes. In the Arctic, the seasonality of SEB fluxes is largely governed by the seasonal changes in incoming radiation, temperature, cloud cover and associated melt and onset of snow cover[15,38]. To assess the role of vegetation type for the seasonal change in SEB-flux magnitudes, we quantify the timing of SEB-flux 'summer-regimes' relative to the timing of the snow-free date in spring and the snow-onset date in autumn[39,40]. Specifically, we characterize the typical seasonal development of mean daily surface energy fluxes for different vegetation types and glacier sites (Fig. 2, Supplementary Fig. 6) using a subset of the data covering the recent two decades (Methods). We define the summer regime for $R_{net}$, H, G, and $T_{surf}$ as the time period when daily mean values exceed 0 W m$^{-2}$ and 0 °C, respectively. For albedo, we define the summer-regime as the time period when daily mean values are below the mean of the yearly minimum and maximum (Methods). Finally, we compare the timing of the summer-regime period of $R_{net}$, H, G, $T_{surf}$ and albedo to the timing of MODIS[41]-observed spring snow-free date and autumn snow-onset date aggregated for each vegetation type (Fig. 3).

We find that for $R_{net}$, H and G, across vegetation types, transitions into the summer-regime occur significantly earlier than the spring

**Table 1 | Estimated mean ± 95% confidence interval (CI) for fluxes of the surface energy budget (SEB) and vegetation type effect significance across summer (JJA) and the annual timescale (Y)**

| Season | Surface energy flux | Mean estimate ± 95% CI for surface energy flux magnitudes in W m$^{-2}$ for each vegetation type (Mean estimate ± 95% CI for normalized magnitudes in % of maximum potential incoming shortwave radiation) | | | | | | | | Effect significance |
|---|---|---|---|---|---|---|---|---|---|---|
| | | Boreal peat bog | Wetland complex | Graminoid | Erect-shrub | Prostrate-shrub | Barren complex | Glacier | | Vegetation type |
| JJA | Rnet | 135 ± 37 (17 ± 5%) | 103 ± 35 (14 ± 4%) | 100 ± 34 (13 ± 4%) | 102 ± 41 (13 ± 5%) | 91 ± 44 (13 ± 6%) | - | 51 ± 49 (6 ± 6%) | | $F_{5,23}$ = 2.1 n.s. |
| | H | 7 ± 40 (1 ± 6%) | 34 ± 36 (5 ± 5%) | 26 ± 36 (3 ± 5%) | 38 ± 37 (5 ± 5%) | 39 ± 36 (6 ± 5%) | −18 ± 32 (−2 ± 5%) | −24 ± 18 (−3 ± 3%) | | $F_{6,20}$ = 3.6* |
| | LE | 75 ± 26 (9 ± 3%) | 32 ± 22 (5 ± 2%) | 47 ± 23 (5 ± 2%) | 25 ± 26 (4 ± 3%) | 24 ± 23 (4 ± 2%) | 0 ± 20 (0 ± 2%) | 8 ± 15 (1 ± 2%) | | $F_{6,10}$ = 5.2* |
| | G | 5 ± 6 (1 ± 1%) | 11 ± 7 (2 ± 1%) | 11 ± 6 (1 ± 1%) | 16 ± 7 (2 ± 1%) | 11 ± 8 (1 ± 1%) | - | 5 ± 7 (1 ± 1%) | | $F_{5,5}$ = 2.4 n.s. |
| | SWnet | 190 ± 36 (23 ± 5%) | 156 ± 34 (21 ± 4%) | 154 ± 32 (20 ± 4%) | 137 ± 45 (18 ± 6%) | 154 ± 41 (21 ± 5%) | - | 87 ± 49 (10 ± 7%) | | $F_{5,22}$ = 3.3* |
| | LWnet | −51 ± 19 (−6 ± 3%) | −39 ± 19 (−5 ± 3%) | −49 ± 18 (−6 ± 3%) | −29 ± 21 (−4 ± 3%) | −38 ± 24 (−6 ± 3%) | −71 ± 17 (−9 ± 2%) | −36 ± 16 (−4 ± 2%) | | $F_{6,15}$ = 5.5** |
| Y | Rnet | 31 ± 17 (3 ± 2%) | 24 ± 19 (1 ± 3%) | 30 ± 19 (5 ± 2%) | - | 11 ± 21 (0 ± 3%) | - | 4 ± 20 (0 ± 2%) | | $F_{4,13}$ = 3.8* |
| | H | - | −1 ± 31 | - | - | −8 ± 31 | −13 ± 22 | −25 ± 13 | | - |
| | LE | - | - | - | - | - | - | - | | - |
| | G | 1 ± 2 (0 ± 0%) | 0 ± 2 (0 ± 0%) | 1 ± 2 (0 ± 0%) | 1 ± 3 | 1 ± 2 (0 ± 0%) | - | −1 ± 2 (0 ± 0%) | | $F_{5,-3}$ = 0.0. n.s. |
| | SWnet | 63 ± 23 (9 ± 5%) | 60 ± 23 (8 ± 7%) | 55 ± 21 (9 ± 7%) | - | 54 ± 23 (7 ± 7%) | - | 34 ± 27 (4 ± 5%) | | $F_{4,10}$ = 1.3 n.s. |
| | LWnet | −35 ± 14 (−6 ± 3%) | −29 ± 14 (−6 ± 3%) | −27 ± 14 (−5 ± 3%) | −25 ± 17 | −32 ± 17 | −40 ± 12 (−7 ± 2%) | −31 ± 10 (−5 ± 2%) | | $F_{6,13}$ = 1.2 n.s. |

Mean estimates ± 95% CI (W m$^{-2}$) are derived from a mixed-model analysis with seasonally aggregated SEB-data (nr. of sites: 64, nr. of site years: 652, period: 1994–2021). Surface energy fluxes: $Rnet$ net radiation, $H$ sensible heat flux, $G$ ground heat flux, $SWnet$ net shortwave radiation, $LWnet$ net longwave radiation. Effect significance is derived from ANOVA (F-value, P-value: ***P < 0.001, **P < 0.01, *P < 0.05, n.s. not significant). Estimates for normalized surface energy fluxes (i.e., percentage of maximum potential incoming shortwave radiation) are indicated in parentheses. Note: flux direction convention is positive away from the surface for heat fluxes (i.e. H, LE and G). Effect significance is not shown where 3 or more vegetation types are missing. See Supplementary Table 4 for significant pairwise differences among vegetation types and Supplementary Table 5 for a more detailed analysis of normalized surface energy fluxes across the summer months.

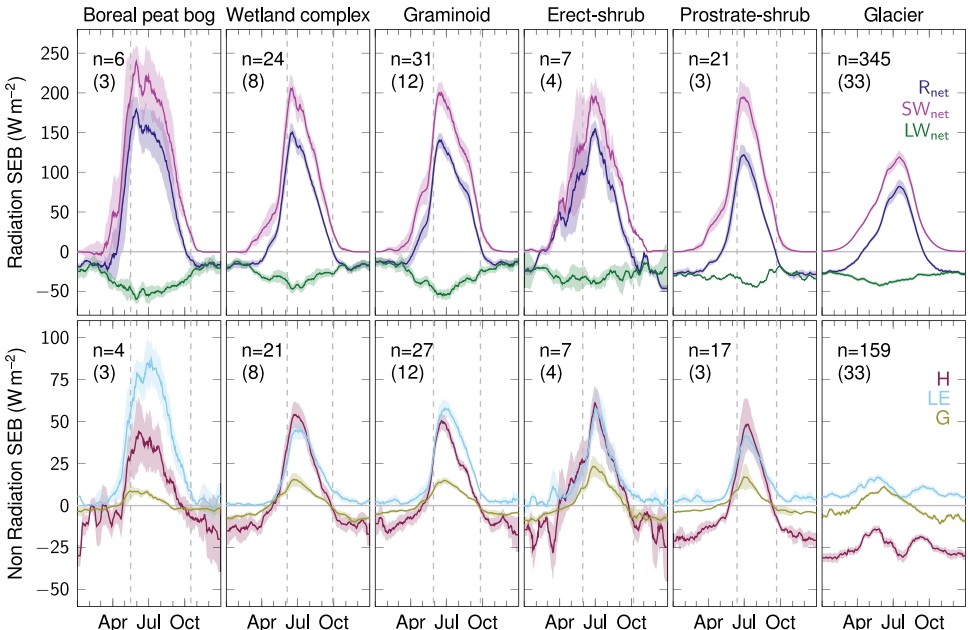

**Fig. 2 | Seasonalities of radiation and non-radiation fluxes of the surface energy budget (SEB).** Surface energy flux values (Wm⁻²) were averaged for each study site for each day of year (DOY) across all years available and then averaged (mean ± s.e.) for each DOY and smoothed (15-day moving average) for each vegetation type. Average number of site years (n) across all DOY's and surface energy fluxes, and the number of study sites (in parentheses) are indicated in the top left corner of each figure. The area within the vertical gray lines represents the median snow-free period across the years 2000–2020 (MODIS[41]), averaged across sites for each vegetation type. Radiation surface energy fluxes: $R_{net}$ (dark blue): net radiation;

$SW_{net}$ (purple): net shortwave radiation; $LW_{net}$ (green): net longwave radiation. Non radiation surface energy fluxes: H (dark red): sensible heat flux; LE (light blue): latent heat flux; G (yellow green): ground heat flux. Results are based on the data subset of the period 2000–2021 and excluding barren vegetation type because of missing $R_{net}$ data: nr. sites = 61, nr. site years = 617. See Supplementary Fig. 6 for seasonality analyses with additional components of the surface energy budget. Note: flux direction convention is positive away from the surface for heat fluxes (i.e. H, LE and G). Source data are provided as a Source Data file.

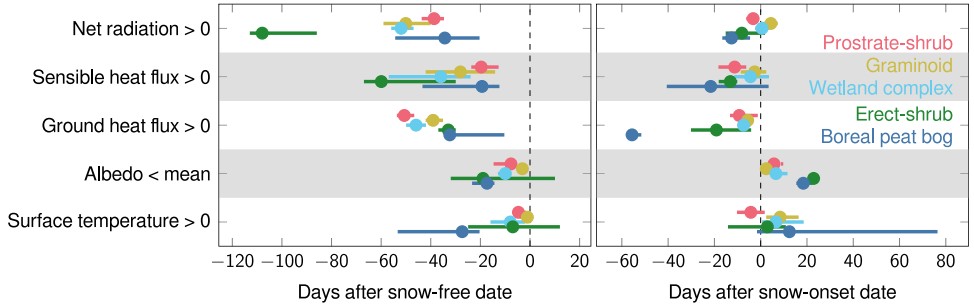

**Fig. 3 | Timing of the summer-regime relative to the snow-free period for selected fluxes cases and components of the surface energy budget (SEB).** Number of days difference between the start of the summer-regime period and the spring snow-free date (left panel, *x*-axis) as well as between the end of the summer-regime period and the autumn snow-onset date (right panel, *x*-axis). The summer-regime timings are derived from the smoothed seasonalities (mean ± s.e., see Fig. 2) of selected surface energy fluxes (*y*-axis), for different vegetation types (*n* = 5 per surface energy flux), colored dots: prostrate-shrub tundra (light red), graminoid tundra (yellow green), wetland complex (light blue), erect shrub tundra (green),

boreal peat bog (blue). We excluded latent heat fluxes since they are >0 Wm⁻² all year in most cases. Summer-regime is defined as the time when surface energy fluxes: >0 Wm⁻², when surface temperature >0 °C, or when albedo <mean of annual minimum and maximum value, respectively (Methods). Results are based on the vegetation subset of the data for the period 2000–2021 and excluding barren vegetation type because of missing net radiation data; nr. sites = 28, nr. site years = 217. Note: flux direction convention is positive away from the surface for heat fluxes (i.e. H, LE and G). Source data are provided as a Source Data file.

snow-free date (mean day of year ± s.d = 151 ± 10; difference in days to the snow-free date for $R_{net}$: −56 ± 27; H: −33 ± 18; G: −39 ± 11; Welch *t*-test *P* < 0.05 for all). The start into the summer-regime for $T_{surf}$ and albedo is not significantly different from the snow-free date (Fig. 3). The timing of the end of the summer-regime is not significantly different from the snow-onset date (mean day of year ± s.d. = 275 ± 10) for all surface energy fluxes, except G, which drops below 0 Wm⁻² before snow cover onset (difference in days to the snow-onset date for $R_{net}$: −3 ± 7; H: −10 ± 11; $T_{surf}$: 8 ± 21; albedo: 12 ± 8; G: −19 ± 20; Welch *t*-test *P* < 0.05 for G).

For $R_{net}$ and H, variability among vegetation types is larger for the start of summer-regime timings compared to the end. Standard deviations for G show the opposite pattern (i.e., are small at the start of season and large at end of season; Fig. 3). For H, the range in start of summer-regime timing relative to the snow-free date is 41 days, with the earliest occurring in erect-shrub tundra (60 days before the snow-free date) and the latest in prostrate-shrub tundra and boreal peat bogs (20 and 19 days before the snow-free date, respectively). An even larger range of 74 days is estimated for the timing in $R_{net}$, which occurs earliest in erect-shrub tundra (108 days before the snow-free date) and

latest in boreal peat bogs (34 days before the snow-free date). By comparison, the ranges in end-timings across vegetation types are 19 days for H and 17 days for $R_{net}$. For G, the range in relative start of summer-regime timing is 19 days (earliest in prostrate-shrub tundra: 51 days before the snow-free date; latest in boreal peat bogs and erect-shrub tundra: 32 and 33 days before the snow-free date, respectively), compared to a 50-day range in end of summer-timing (earliest in boreal peat bogs: 56 days before the snow-onset date; latest in graminoid tundra: 6 days before the snow-onset date).

## Discussion

Our quantitative assessment of surface energy budget (SEB) observations across the treeless, terrestrial Arctic shows that vegetation type is a powerful predictor of summer (JJA) surface energy fluxes, especially for latent and sensible heat fluxes. The results remain robust for normalized surface energy fluxes that are adjusted for potential incoming shortwave radiation and indicate that differences among vegetation types can be as significant as differences between glaciated and vegetated surfaces.

One reason for the strong predictive power of vegetation type could be that vegetation types reflect integrative proxies of distinct combinations of environmental conditions that control the SEB, including temperature, topography, soil moisture, and permafrost characteristics[18–20,26]. However, it has also been shown that the CAVM classes[20], upon which our vegetation types are largely based, do differ in SEB-relevant traits and functions such as vegetation height, productivity and albedo[25,42], and therefore, causal effects of vegetation types can also be expected. We argue that it is likely a combination of both reasons behind the observed predictive power of vegetation type. Further research will be needed to identify the specific vegetation trait combinations that are most relevant for the SEB in the Arctic, including traits not represented by the main CAVM classes we used (Supplementary Discussion), and to better constrain the implications of changing Arctic vegetation productivity, plant height and shrub abundance for the SEB[6–8]. Nevertheless, our results highlight the potential for improving the predictions of Arctic surface energy fluxes, specifically of summer latent and sensible heat fluxes, by a more comprehensive treatment of vegetation types and how environmental conditions interact with associated vegetation functional traits[18,19].

Our variance partitioning analysis also reveals a low predictive power of cloud cover, which is counter-intuitive[15,38], but can be explained by the persistence of high fractional cloud cover and its consequent low spatial variability across large parts of the Arctic in summer[43]. Therefore, even though cloud cover has a large effect on the overall surface radiative forcing, it does not explain much variability among our study sites[15,38,43]. The relatively small predictive power of precipitation could be indicative of energy-limited, rather than water-limited systems[44] or be due to the fact that the precipitation regime is already captured by other variables in our analysis, including vegetation type[11].

Our mixed-model analyses indicate significant statistical relationships of vegetation type with summer surface energy flux magnitudes (Table 1) that are robust across individual summer months for sensible and latent heat fluxes (Supplementary Table 5, analysis with latitude-adjusted surface energy fluxes). Interestingly, surface energy flux differences among vegetation types can be as high or even higher than among glacier and vegetation sites, especially in the case of sensible and latent heat fluxes (Supplementary Table 4).

For example, in the case of latent heat flux, significant absolute differences between glacier and vegetation types are in a similar range (38.5–66.8 Wm$^{-2}$) as differences among vegetation types only (range: 28.3–74.9 Wm$^{-2}$; Supplementary Table 4). Highest summer latent heat fluxes are found in boreal peat bogs where they are about twice as high as in Arctic wetlands and shrub-dominated tundra (Table 1), all of which show sensible heat fluxes of equal or greater magnitude than

latent heat fluxes[21,45]. Boreal peat bogs are characterized by high moss coverage lacking stomatal control of water vapor exchange with the atmosphere[46,47]. Future changes in peatland cover depend on complex vegetation-water cycle feedbacks that additionally can be mediated by permafrost thaw and microtopography[45–47]. These feedbacks remain poorly understood, partially because most current coupled Earth system models lack a representation of peatland functional types[46]. Nevertheless, in case that boreal peatlands were to expand further north into the Arctic tundra with climate warming[18], our results suggest a potential shift of surface energy partitioning towards latent heat flux, with implications for current and future evaporative water loss, precipitation and soil water availability[46].

In the case of summer sensible heat flux, differences between barren and shrub-dominated tundra are of similar magnitude to differences between glacier and other vegetation types (Supplementary Table 4). Interestingly, barren tundra, similar to glaciers but in contrast to all other vegetation types, has a negative summer sensible heat flux, indicating that barren tundra areas are currently a heat sink. Barren tundra covers around one-fifth of the terrestrial Arctic area[20] and is expected to decrease under Arctic climate warming[6], which suggests a potential positive feedback to climate warming. However, barren tundra is only covered by one study site in our in situ observations dataset and generally underrepresented in the surface energy budget literature (Supplementary Discussion). Therefore, our results need to be interpreted with caution and further research is needed to assess potential climate feedbacks with more certainty.

In this study, we synthesize observations on the seasonality of surface energy fluxes for a consistent classification of vegetation types across the Arctic. Our results indicate that vegetation types differ in the seasonal timing of surface energy flux summer-regime start and end, relative to the spring snow-free and the autumn snow-onset date, respectively. The timing of snow-cover disappearance in spring and onset in autumn is a major factor controlling both the SEB[15,38,48] and vegetation activity[9,49] in the Arctic. However, vegetation type can also affect the distribution, trapping and density of snow cover, with important consequences for snowmelt and snow onset[17,39,40], as well as for carbon[35,50,51], water[17,52] and energy fluxes[17,52]. Our results suggest elevated variability among vegetation types in the estimated start (and end) dates of net positive $R_{net}$ and H (and G) relative to the spring snow-free (and autumn snow-onset) dates. For example, we show an estimated start of net positive H relative to the snow-free date that occurs 40 days earlier in erect-shrub tundra compared to prostrate-shrub tundra, which could be indicative of erect-shrub vegetation protruding through the snow cover earlier compared to prostrate-shrub vegetation[40]. While erect-shrub and prostrate-shrub tundra show similar average summer magnitudes in H (Table 1), these temporal differences can have implications for the cumulative sensible energy flux to the atmosphere[48]. The nature of multiple interactions among vegetation types and snow cover is complex and trends in the timing of snow-free and -onset dates, as well as vegetation phenology are highly heterogeneous across the Arctic[9,10,49]. Resolving related consequences for surface energy fluxes, the soil thermal regime and permafrost thaw need further and more detailed investigations[7,15,17]. We contend that in this effort, the seasonality of surface energy fluxes is as important as their magnitude[48].

Our study highlights important data gaps. Long-term, year-round SEB data series of Arctic vegetation are still very scarce. Data is often missing in autumn and winter seasons for many sites (Supplementary Discussion) and therefore, we could resolve SEB-seasonalities only at the vegetation-type level (and not study-site level; Methods). Furthermore, year-round turbulent flux measurements (i.e. sensible and latent heat flux) are especially scarce for all vegetation types[53]. Finally, SEB observations for barren tundra are largely missing, while this type shows largest differences in surface energy fluxes to other tundra vegetation types in the limited data available for this study.

To conclude, changes in the surface energy budget[12,13] lie at the heart of changes in climate[1,4,5] that can affect the composition, structure and function of Arctic vegetation[3,6–10]. The Arctic system is highly sensitive to climate change[3,18], exerts key land-feedbacks relevant for global climate dynamics[2,54] and harbors a range of vegetation types with unique traits including mosses and lichens[18–20]. For the future, a widespread redistribution of Arctic vegetation is predicted[6,55,56]. Understanding and predicting how these changes in turn affect climate is essential for reducing persisting uncertainties in climate projections[2,16,18]. The potentially important but still uncertain role of Arctic vegetation for climate feedbacks has been highlighted before[18,19]. Previous studies have shown that local estimates of surface energy fluxes are improved if land surface components of Earth system models account for differentiated Arctic PFTs at several high-latitude validation sites and -regions[18,47,57]. Here, we provide quantitative evidence of the importance of vegetation types for predicting Arctic surface energy budgets at circumpolar scale and support recent calls for refined accounting of high-latitude vegetation types and associated vegetation functions in land surface components of Earth system models[18,19,58].

## Methods

### Surface energy fluxes and components

In our study, we focused on the circumpolar land north of 60° latitude, and specifically on the extent of the circumpolar Arctic vegetation map (CAVM[20], Supplementary Fig. 1–3). We obtained half-hourly and hourly in situ observations of energy fluxes and meteorological variables from the monitoring networks FLUXNET[28] (fluxnet.org; FLUXNET2015 dataset), AmeriFlux[29] (ameriflux.lbl.gov), AON[31,32] (aon.iab.uaf.edu), ICOS (icos-cp.eu), GEM[35,36] (g-e-m.dk), GC-Net[33,34] (cires1.colorado.edu/steffen/gcnet) and PROMICE[30]; (promice.dk; Supplementary Table 3). We did not include observations from the Baseline Surface Radiation Network (BSRN; bsrn.awi.de) and Global Energy Balance Archive (GEBA; geba.ethz.ch) because they typically lack information on non-radiative energy fluxes. Finally, we did not include observations from the European Flux Database Cluster (EFDC, europe-fluxdata.eu) because these data are largely located outside the domain of the CAVM[20].

We aggregated surface energy fluxes and components (Supplementary Table 1) to daily resolution as follows: (i) we extracted only directly measured data and excluded gap-filled data by filtering according to quality information; (ii) we performed a basic outlier filtering (excluding shortwave and longwave radiation flux values >1400 Wm$^{-2}$ and in case of incoming/outgoing radiation <0 Wm$^{-2}$, excluding albedo values <0 and >1, excluding air and surface temperatures < −100 °C; (iii) we converted all times of measurements to local standard time (i.e. without daylight saving time); (iv) we calculated daily (i.e. 24 h) mean, minimum and maximum values for all days and surface energy fluxes and components where a minimum of 65% of data was available with a maximum temporal gap of 4.8 h; (v) we extracted metadata (units, measurement heights, original variable names, instrumentation) for all sites and variables; (vi) when data for the same time and location were available from several networks (partial overlapping of FLUXNET and AON with Ameriflux), we averaged respective daily values across networks; (vii) in case data for the same time and location were measured by several sensors at one site (replicated measurements), we averaged across replicates; (viii) we harmonized units for all variables; (ix) we harmonized the flux direction convention for non-radiative energy fluxes (i.e. H, LE, and G; Supplementary Table 1) as „positive away from surface"; (x) we derived net radiation (R$_{net}$), net shortwave radiation (SW$_{net}$), net longwave radiation (LW$_{net}$) and albedo from corresponding daily aggregated incoming and outgoing shortwave and longwave radiation, respectively, if not otherwise available; (xi) we derived normalized fluxes for R$_{net}$ (n.R$_{net}$), SW$_{net}$ (n.SW$_{net}$), LW$_{net}$ (n.LW$_{net}$), H (n.H), LE (n.LE) and G

(n.G) as the percentage (%) of daily maximum potential incoming shortwave radiation based on location and topographical conditions[59]. The final surface energy budget dataset (named SEB-data hereafter) consists of daily mean, minimum and maximum values for surface energy fluxes and components for 64 tundra and glacier sites across the years 1994–2021 (Supplementary Figs. 1–3; Supplementary Table 2; nr. sites = 64, nr. site years = 652).

### Drivers of the surface energy budget (SEB-drivers)

We characterized all study sites available in the SEB-data according to environmental covariates (named SEB-drivers hereafter; Supplementary Table 1) that potentially have important effects on the magnitude and seasonality of surface energy fluxes from ancillary geographic data (Supplementary Fig. 4, Supplementary Table 3). We identified these relevant SEB-drivers according to our literature synthesis and consequent correlation analysis (Supplementary Discussion, Supplementary Figs. 7–9).

As the main SEB-driver of interest, we focused on the local-scale, in situ vegetation type (Vegetation type variable), which reflects the main classes described in the CAVM[20] map (barren complexes, graminoid tundra, prostrate dwarf-shrub tundra, erect-shrub tundra, wetland complexes, glacier), plus boreal peat bog.

To derive this in situ vegetation type variable, we extracted for each study site in the SEB-data the vegetation descriptions from adequate literature references (Supplementary Table 2). We categorized each site's vegetation according to the most adequate CAVM-class by using the following decision chain: (i) assign the CAVM-class where most described species from in situ vegetation descriptions and Table S1 in ref. 20 are in accordance. Take into account species' ecological niche sizes (e.g. stenotopic plants are better indicators than eurytopic plants); (ii) if there is no species list and/or in case there are several CAVM candidate classes, take into account the general habitat and ecosystem descriptions (including community vertical and horizontal structure, soil moisture and acidity, topography); (iii) if there are still several CAVM candidate classes, take into account the bioclimatic subzone and evaluate the exact location of the site with available satellite data; (iv) if there are still several CAVM candidate classes, assign the most dominant CAVM-class; (v) in case of uncertainties, consult with the study author(s); (vi) If the vegetation is not similar to any CAVM-class, describe it according to the description of the study author(s) and/or adequate references (Supplementary Table 2).

This categorization revealed 1 barren tundra site (1 B3), 12 graminoid tundra sites (10 G4, 1 G3 and 1 G1), 3 prostrate-shrub tundra sites (2 P2 and 1 P1), 4 erect-shrub tundra sites (2 S2 and 2 S1), 8 wetland sites (3 W3, 3 W2 and 2 W1), 33 glacier sites (GL), and three sites that were not similar to any CAVM-class but clearly identified as boreal peat bogs (Supplementary Figs. 1–3; Supplementary Table 2).

For each study site in the SEB-data, we extracted the landscape-scale, dominant vegetation type in the site-surrounding area (radius = 500 m, CAVM type variable) from the raster version of the CAVM[20,60] (Supplementary Fig. 4a; Supplementary Table 3). This map is a refined version of the widely used original version[20,60] and provides information at 1 km$^2$ spatial resolution and pan-Arctic scale north of the Arctic treeline[20]. Compared to the Vegetation type variable, the CAVM type is based on the analysis of a combination of large-scale satellite and environmental data, but both variables refer to the same class definition. The CAVM-classes are based on the plant physiognomy of the zonal vegetation in a given area, analogous to the widely applied Braun-Blanquet approach for plant communities on the ground[20].

The CAVM types used in this study reflect these main plant physiognomies (plus one glacier class) that are named according to the following dominant plant growth forms: B: barren and barren complexes (4 sites), G: graminoid tundra (10 sites), P: prostrate dwarf-shrub tundra (1 site), S: erect-shrub tundra (10 sites), W: wetland complexes

(3 sites), GL: glacier (30 sites; there are 6 non-Arctic sites in our SEB-data). A confusion matrix showing the classification of study sites into Vegetation type and CAVM type categories is shown in Supplementary Table 6.

For each study site, we averaged the mean annual air temperature (°C), the annual sum of precipitation (mm) and the annual sum of snow water equivalent (Snow amount variable, mm; defined as daily precipitation when daily mean air temperature < =0 °C) across the years 1979-2018. Therefore, we used high resolution downscaled model output estimates of temperature and precipitation (CHELSA V2.1; „tas" and „pr" variables[61–63]; Supplementary Fig. 4d, e; Supplementary Table 3).

Using the same air temperature data as above[61,63], for each study site we calculated the average Conrad Continentality Index (CCI[64]) across the years 1979–2018. Specifically, we used the formula:

$$\text{Continentality} = \frac{1.7 \times (T\max - T\min)}{\sin(\varphi + 10)} - 14 \quad (2)$$

whereby $T_{max}$ and $T_{min}$ (°C) refer to the mean air temperature of the warmest and coldest month, respectively, and $\varphi$ (radians) refers to latitude.

Using the same air temperature data as above[61,63], for each study site we calculated the average Summer Warmth Index (i.e. the annual sum of monthly mean air temperatures above 0 °C; SWI[42]) across the years 1979–2018.

For each study site, we extracted the bioclimatic subzone (CAVM subzone variable) as described in the circumpolar Arctic vegetation map (CAVM[20,60]). These five bioclimatic subzones (A-E) and additional classes for glacier and non-Arctic zones are largely based on a combination of Arctic phytogeographic zones[65], dominant growth forms of plants and summer temperatures[66]. Hence, these bioclimatic subzones are generally well aligned with summer warmth index (SWI) classes[42]. A confusion matrix showing the classification of study sites into Vegetation type and CAVM subzone categories is shown in Supplementary Table 7.

For each study site, we extracted the median snow cover duration (Snow duration variable) from satellite-sensed daily snow cover (MODIS MOD10C1[41]; Supplementary Table 3). Snow duration was calculated for each year as the number of days between the spring snow-free date (i.e. mean of days of year for last ‚snow' day and first "no snow" day MODIS categories) and the autumn snow-onset date (i.e. mean of days of year for last "no snow" day and first "snow" day MODIS categories). Yearly snow duration, snow-free date and snow-onset date for each study site were then aggregated by calculating the median across the years 2000–2020.

For each study site, we extracted mean annual cloud cover (%) and cloud-top temperature (°C) from monthly satellite imaging radiometer data ("cldamt" and "tc" products; ISCCP-Basic-H series[67,68]; Supplementary Fig. 4f, g; Supplementary Table 3) and averaged these variables for each study site across the years 1984–2016.

For each study site we extracted the corresponding permafrost extent as described in the Circum-Arctic permafrost and ground ice map[69] (NSIDC gdd318_map_circumArctic version 2; Supplementary Fig. 4b; Supplementary Table 3). This map describes 4 categories of permafrost extent, based on the percentage of the ground that is underlain by permafrost. There are additionally 5 separate categories for glaciers, relict permafrost, inland lakes, oceans and land with no permafrost. For the Permafrost extent variable used in this study, we aggregated these categories into the following five classes: continuous permafrost (C; 90–100% extent; 21 sites); discontinuous permafrost (D; 50–90% extent; 5 sites); sporadic or isolated patches of permafrost (Si; <50% extent; 4 sites), ocean/inland seas (o; 2 sites) and glaciers (g; 32 sites). A confusion matrix showing the classification of study sites

into Vegetation type and Permafrost extent categories is shown in Supplementary Table 8.

Using the same permafrost data[69] (Supplementary Fig. 4b; Supplementary Table 3), for each study site we extracted the corresponding permafrost ground-ice content class: high (h; >20% ice content; 8 sites), medium (m; 10–20% ice content; 7 sites), low (l; 0–10% ice content; 15 sites), ocean/inland seas (o; 2 sites), and glaciers (g; 32 sites). A confusion matrix showing the classification of study sites into Vegetation type and Permafrost ice content categories is shown in Supplementary Table 9.

For each study site we extracted the average altitude (m a.s.l.), slope (°) and northness of the aspect (1 if north-exposed, −1 if south-exposed; derived from the cosine of aspect in radians) in the surrounding area (radius = 500 m) from the satellite-sensed digital elevation model raster mosaic at 100 m spatial resolution (ArcticDEM: Arcticdem_mosaic_100 m_v3.0; ref. 70; Supplementary Fig. 4c; Supplementary Table 3).

### Data analysis

For our analyses, we focused on the surface energy fluxes and components net radiation ($R_{net}$, Wm$^{-2}$), sensible heat flux (H, Wm$^{-2}$), latent heat flux (LE, Wm$^{-2}$), ground heat flux (G, Wm$^{-2}$), net shortwave radiation ($SW_{net}$, Wm$^{-2}$), net longwave radiation ($LW_{net}$, Wm$^{-2}$), albedo (unitless), and surface temperature ($T_{surf}$, °C), air temperature ($T_{air}$, °C) and the difference between surface and air temperature ($T_{surf}$-$T_{air}$; Supplementary Table 1). We included albedo and $T_{surf}$ because they are directly related to the radiative SEB as follows[14]:

$$R_{net} = SW_{in}(1 - \text{albedo}) + LW_{in} - \varepsilon\sigma T_{surf}^4 \quad (3)$$

where $SW_{in}$ and $LW_{in}$ are the incoming shortwave and longwave radiation, respectively (Wm$^{-2}$), $\varepsilon$ is the surface emissivity ($\cong$1) and $\sigma$ is the Stefan-Boltzmann constant[14]. We repeated our analyses for normalized fluxes of $R_{net}$ (n.$R_{net}$), $SW_{net}$ (n.$SW_{net}$), $LW_{net}$ (n.$LW_{net}$), H (n.H), LE (n.LE) and G (n.G), which are all expressed in percent of maximum potential incoming shortwave radiation[59] (%; Supplementary Table 1).

We conducted all data processing and analyses using the R-software version 3.6.0[71]. We analyzed the full SEB-dataset (nr. sites = 64, nr. site years = 652) and separately a "vegetation" data subset excluding glacier sites (nr. of sites: 31, nr. of site years: 234). Furthermore, we conducted all analyses at the yearly timescale and for the summer season (JJA; June, July, August), which is when in situ observations for vegetation sites have more complete data coverage (Supplementary Discussion). Summer is also the time when the SEB is dominated by absorbed solar radiation, and local controls by the land surface are expected to be more important than in the winter season[27,38]. In the winter season, absorbed solar radiation is negligible and the SEB is largely influenced by synoptic processes, such as advection from lower latitudes[27,38]. We chose to use the standard meteorological summer season (as opposed to "snow-covered" vs. "snow-free" season) because (1) the JJA season is largely snow free for relevant cases (i.e. except glacier sites), and (2) standard seasons are consistently defined in time; reducing confounding of results due to seasonal changes in solar irradiance.

We compared the relative importance of SEB-drivers for explaining variance in the surface energy fluxes and components $R_{net}$, H, LE, G, $SW_{net}$, $LW_{net}$, albedo and $T_{surf}$. We repeated this analysis for normalized fluxes n.$R_{net}$, n.$SW_{net}$, n.$LW_{net}$, n.H, n.LE and n.G (Fig. 1 and Supplementary Fig. 5). Specifically, we focused on the importance of the SEB-driver "Vegetation type" compared with the importance of other drivers related to landscape-scale dominant vegetation type (CAVM type), climate (Temperature, Precipitation, Snow amount, Continentality, Summer warmth, CAVM subzone), clouds (Cloud cover, Cloud temperature), snow (Snow duration), permafrost (Permafrost

extent, Permafrost ice content), topography (Altitude) and geographic location (Latitude). These 15 SEB-drivers were selected based on our previous literature synthesis and consequent correlation analysis (Supplementary Discussion, Supplementary Figs. 7–9).

Using the "vegetation" data subset excluding glacier sites (nr. of sites: 31, nr. of site years: 234), we averaged surface energy fluxes for each summer season (JJA), year and study site, where at least 80% of daily measurements were available. To calculate the relative importance for each SEB-driver, we applied a variance partitioning method[72,73] to predict surface energy flux values averaged for each study site. Specifically, we used the set of 15 selected SEB-drivers to build all possible models with 2 predictors (a number high enough to allow for the pairwise assessment of statistical confounding among predictors and low enough to avoid model overfitting). For each model, we quantified the variance explained by each predictor in a predictor pair when fitted first, when fitted last and when averaged over all possible orderings in the models. Finally, we averaged the "first", "last" and "average" explained variance (%) for each SEB-driver across all models for each surface energy flux variable. This led to the testing of 105 unique SEB-driver pairs × 2 predictor orders × 14 surface energy flux variables = 2940 models for the vegetation data subset.

We estimated the magnitude of the surface energy fluxes $R_{net}$, $SW_{net}$, $LW_{net}$, H, LE and G for each Vegetation type at seasonal (JJA) and yearly timescale (Y) using the full SEB-dataset (nr. sites = 64, nr. site years = 652; Table 1; Supplementary Table 4). We repeated this analysis for the monthly aggregated, normalized (i.e. potential incoming radiation-adjusted[59]) summer fluxes $n.R_{net}$, $n.SW_{net}$, $n.LW_{net}$, n.H, n.LE, n.G, albedo and $T_{surf}$ using the vegetation data subset (excluding glacier sites; nr. of sites: 31, nr. of site years: 234; Supplementary Table 5).

In both analyses, we averaged the daily mean values of the surface energy fluxes for each timescale, year and study site, where at least 80% of days with measurements were available. We then averaged surface energy flux values across years for each study site and estimated the mean ± 95% confidence interval (CI) as a function of Vegetation type by using a linear mixed-model analysis. Specifically, we modeled the study-site aggregated means of each surface energy flux as a function of Vegetation type (fixed effect) and the corresponding data distribution network (i.e. Ameriflux, FLUXNET etc.; random effect; Supplementary Table 2). To compare differences of summer surface energy flux estimates among Vegetation types, we applied a consequent post-hoc pairwise comparison with bonferroni correction of significance estimates[74] (Supplementary Table 4).

We derived the typical seasonal change of the surface energy fluxes $R_{net}$, $SW_{net}$, $LW_{net}$, H, LE and G for each Vegetation type (Fig. 2). We repeated this analysis for the normalized (i.e. maximum potential incoming radiation-adjusted[59]) fluxes $n.R_{net}$, $n.SW_{net}$, $n.LW_{net}$, n.H, n.LE, n.G, albedo, $T_{surf}$, $T_{air}$ and $T_{surf}$-$T_{air}$ (Supplementary Fig. 6).

Specifically, using data constrained to the years 2000–2021 and excluding barren vegetation type (B; because of missing Rnet data; nr. of sites = 61, nr. site years = 617), we averaged the daily mean values of the surface energy fluxes across all available years for each day of year (DOY) and study site. In a second step, we grouped study sites by Vegetation type and derived mean ± s.e. for each DOY and surface energy flux variable. Finally, we smoothed the resulting mean ± s.e. values for each Vegetation type by calculating the centered 15 day moving average for each DOY (Fig. 2).

We used these smoothed seasonalities for the five Vegetation types Boreal peat bog, Wetland complex, Graminoid tundra, Erectshrub tundra, Prostrate-shrub tundra (excluding Glacier; nr. of sites = 28, nr. site years = 217), to assess the start and the end of the "summerregime" period for $R_{net}$, H, G, albedo and $T_{surf}$. In the case of $R_{net}$, H, G and $T_{surf}$, we defined the summer-regime as the time period where values exceed 0 Wm$^{-2}$ and 0 °C, respectively. In the case of albedo, we defined the summer-regime as the period when values fall below the mean of the yearly minimum and maximum value[75,76]. We excluded LE

since fluxes are >0 Wm$^{-2}$ all year in most cases. Using this data, we compared the timing of the summer regime of $R_{net}$, H, G, $T_{surf}$ and albedo to the timing of the snow-free period (Snow duration[41]) for each Vegetation type. Specifically, we aggregated the spring snow-free and autumn snow-onset dates for each Vegetation type and subtracted these from the start and end of the summer-regime period of the selected surface energy fluxes, respectively. Finally, we aggregated the start and end of summer-regime timings across Vegetation types for each selected surface energy flux variable, to test the differences in the timing of the surface energy flux summer-regime and snow-free/snow-onset dates, using corresponding two-sample Welch's $t$-tests (Fig. 3).

## Data availability

Source data are provided with this paper. The complete in situ observations surface energy budget components dataset (SEB-data), the SEB-driver dataset, as well as the literature synthesis dataset generated in this study have been deposited[77] in the PANGAEA database under accession code: https://doi.pangaea.de/10.1594/PANGAEA.949792. AmeriFlux data can be accessed at: https://ameriflux.lbl.gov/login/?redirect_to=/data/download-data/ AON data can be accessed at: http://aon.iab.uaf.edu/data_access FLUXNET (including GEM and ICOS) data can be accessed at: https://fluxnet.org/data/download-data/ GC-Net data can be accessed at: http://cires1.colorado.edu/science/groups/steffen//gcnet/ PROMICE data can be accessed at: https://promice.org/download-data/ Circumpolar Arctic Vegetation Map data can be accessed at: https://data.mendeley.com/datasets/c4xj5rv6kv/1 Bioclimatic subzones can be accessed at: http://www.arcticatlas.org/maps/themes/cp/cpbz Climate data can be accessed at: https://chelsa-climate.org/ Snow cover data can be accessed at: https://nsidc.org/data/MOD10C1/versions/6#0 Cloud data can be accessed at: https://www.ncei.noaa.gov/data/international-satellite-cloud-climate-project-isccp-h-series-data/access/isccp-basic/ Permafrost data can be accessed at: ftp://sidads.colorado.edu/pub/DATASETS/fgdc/ggd318_map_circumArctic/ Altitude data can be accessed at: http://data.pgc.umn.edu/elev/dem/setsm/ArcticDEM/mosaic/v3.0/100m/ Source data are provided with this paper.

## Code availability

Code underlying this study is available in a Github repository and can be accessed at: https://github.com/oehrij/ArcticSEBSynthesis[78].

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

## Acknowledgements

We acknowledge the funding and the support of the Swiss National Science Foundation SNF Grant Nr. 178753 (http://p3.snf.ch/Project-178753) to G.S.S., the University of Zurich Research Priority Program Global Change and Biodiversity (URPP GCB, https://www.gcb.uzh.ch/en.html) and the EU-CHARTER project (European Commission RIA #869471, http://www.charter-arctic.org/). We thank Colin R. Lloyd, Bruce C. Forbes and John C. Moore for their support and stimulating discussions. Funding for the AmeriFlux data portal was provided by the U.S. Department of Energy Office of Science. Data from the Programme for Monitoring of the Greenland Ice Sheet (PROMICE) and the Greenland Analogue Project (GAP) were provided by the Geological Survey of Denmark and Greenland (GEUS) at http://www.promice.dk. AON datasets were provided by the Institute of Arctic Biology, UAF, based upon work supported by the National Science Foundation under grant #1503912. The National Ecological Observatory Network is a program sponsored by the National Science Foundation and operated under cooperative agreement by Battelle. This material is based in part upon work supported by the National Science Foundation through the NEON Program. ArcticDEM data is provided by the Polar Geospatial Center under NSF-OPP awards 1043681, 1559691, and 1542736. The ISCCP H-series cloud data is provided by NOAA/NCEI. MODIS data is provided by the NASA National Snow and Ice Data Center Distributed Active Archive Center (NSIDC DAAC).

Ameriflux and FLUXNET site ID's and corresponding doi's: CA-SCB (https://doi.org/10.17190/AMF/1498754), FI-Lom (https://doi.org/10.18140/FLX/1440228), GL-NuF (https://doi.org/10.18140/FLX/1440222), GL-ZaF (https://doi.org/10.18140/FLX/1440223), GL-ZaH (https://doi.org/10.18140/FLX/1440224), RU-Che (https://doi.org/10.18140/FLX/1440181), RU-Cok (https://doi.org/10.18140/FLX/1440182), RU-Sam (https://doi.org/10.18140/FLX/1440185), RU-Tks (https://doi.org/10.18140/FLX/1440244), RU-Vrk (https://doi.org/10.18140/FLX/1440245), SE-St1 (https://doi.org/10.18140/FLX/1440187), SJ-Adv (https://doi.org/10.18140/FLX/1440241), SJ-Blv (https://doi.org/10.18140/FLX/1440242), US-A03 (https://doi.org/10.17190/AMF/1498752), US-A10 (https://doi.org/10.17190/AMF/1498753), US-An1 (https://doi.org/10.17190/AMF/1246142), US-An2 (https://doi.org/10.17190/AMF/1246143), US-An3 (https://doi.org/10.17190/AMF/1246144), US-Atq (https://doi.org/10.17190/AMF/1246029), US-Brw (https://doi.org/10.17190/AMF/1246041), US-EML (https://doi.org/10.17190/AMF/1418678), US-HVa (https://doi.org/10.17190/AMF/1246064), US-ICh (https://doi.org/10.17190/AMF/1246133), US-ICs (https://doi.org/10.17190/AMF/1246130), US-ICt (https://doi.org/10.17190/AMF/1246131), US-Ivo (https://doi.org/10.17190/AMF/1246067), US-NGB (https://doi.org/10.17190/AMF/1436326), US-Upa (https://doi.org/10.17190/AMF/1246108), US-xHE (https://doi.org/10.17190/AMF/1617729), US-xTL (https://doi.org/10.17190/AMF/1617739).

This work was supported in part by the Research Network for Geosciences in Berlin and Potsdam (SO 087 GeoX to I.G.), the U.S.

Department of Energy (DE-SC0013306 to G.d.b., DE-AC02-06CH11357 to R.C.S.), the National Science Foundation (#1936752 to M.S.B.-H., E.S.E., C.W.E., #1556772 to A.V.R.), the Swiss National Science Foundation (20BD21_193907 and 20BD21_184131 to D.N.K.), the EU-BASIS programme (ENV4-CT97-0637 to R.J.H.), the German Research Foundation (EXC 2037 'CLICCS' 390683824 to D.H. and L.K.), the National Science Centre, Poland (2017/25/B/ST10/00540 to I.S.), the Research council of Norway (no. 274711 to F.-J.W.P. and no. 301552 to N.P.), the Swedish Research Council (no. 2017-05268 to F.-J.W.P.), Greenland Research Council (no. 80.35 to E.L.-B.), the Arctic Challenge for Sustainability II (JPMXD1420318865 to M.U.), the National Research Foundation of Korea (NRF-2020R1A6A3A03038242 to J.-S.K.), the NOAA Physical Sciences Laboratory, ARM (arm.gov) and ArcticNet (arcticnet.ulaval.ca).

This study is dedicated to Koni Steffen who tragically lost his life on 8 August 2020, while maintaining energy budget measurement instrumentation at Swiss camp on the rapidly melting ice sheet of Greenland.

## Author contributions

Study design, methods and analysis were conceived by: J.Oe., G.S.S., J.-S.K., R.G. Data was contributed by: J.Oe., I.G., J.Bo., B.V., C.W.E., D.N.K., M.J.-K., E.L.-B., M.M., S.Me., K.S., F.-J.W.P., N.P., W.L.Q., T.S., O.S., J.Be., M.S.B.-H., T.R.C., J.F.D., R.S.F., M.G., A.V.R., B.R.K.R., R.C.S., P.C.S., J.A.G., R.J.H., A.K., M.K., M.t-B., T.F., S.N.W., G.d.B., L.D.R., I.S., E.A.G.S, D.H., L.K., J.P., D.v.A., C.W. Data was processed and analyzed by: J.Oe., V.Z., I.G., M.R.C. Article drafting was done by: J.Oe., G.S.S., J.-S.K., R.G., H.K., I.G., O.S., E.S.E. Article figures were done by: J.Oe., I.G. Revising article for intellectual content was done by: J.Oe., G.S.S., J.-S.K., R.G., H.K., I.G., V.Z., O.S., E.S.E., M.R.C., G.M., P.D.B., J.F.D., A.d.S., R.J.H., I.S., L.K., E.P., A.R., J.Bo., N.B.M., J.Be., E.L.-B., P.C.S., R.C.S., M.K., F.-J.W.P., J.A.G., M.M., C.W., M.J.-K., D.N.K., W.L.Q., J.P., D.v.A., T.R.C., M.Z.H., R.S.S., S.Me., B.V., G.V.F., M.W., B.H., D.M., F.D., M.t.B., T.S., A.K., A.V.R., S.N.W., S.Mo., A.L.A., R.E., B.R.K.R., D.H., L.D.R., H.I., E.A.G.S., C.J.C., A.A.G., J.P.M., R.S.F., M.G., M.U., N.P., G.d.B., M.S.B.-H., M.L., K.S., T.F., A.O., C.W.E., J.Ol., S.D.C.

## Competing interests

The authors declare no competing interests.

## Additional information

**Correspondence and requests** for materials should be addressed to Jacqueline Oehri or Gabriela Schaepman-Strub.

Jacqueline Oehri [1,2,65] ✉, Gabriela Schaepman-Strub [1,65] ✉, Jin-Soo Kim [1,3,64], Raleigh Grysko[1,64], Heather Kropp[4,64], Inge Grünberg [5,64], Vitalii Zemlianskii[1,64], Oliver Sonnentag [6,64], Eugénie S. Euskirchen [7,64], Merin Reji Chacko[1,8,9,64], Giovanni Muscari[10], Peter D. Blanken [11], Joshua F. Dean [12], Alcide di Sarra[13], Richard J. Harding[14], Ireneusz Sobota[15], Lars Kutzbach [16], Elena Plekhanova[1], Aku Riihelä [17], Julia Boike [5,18], Nathaniel B. Miller[19], Jason Beringer [20], Efrén López-Blanco [21,22], Paul C. Stoy[19], Ryan C. Sullivan [23], Marek Kejna[24], Frans-Jan W. Parmentier [25,26], John A. Gamon [27], Mikhail Mastepanov[22,28], Christian Wille [29], Marcin Jackowicz-Korczynski[22,26], Dirk N. Karger [30], William L. Quinton[31], Jaakko Putkonen[32], Dirk van As[33], Torben R. Christensen[22,28], Maria Z. Hakuba[34], Robert S. Stone[35], Stefan Metzger [36,37], Baptiste Vandecrux[33], Gerald V. Frost [38], Martin Wild [39], Birger Hansen[40], Daniela Meloni [41], Florent Domine [42,43], Mariska te Beest[44,45], Torsten Sachs [29], Aram Kalhori[29], Adrian V. Rocha[46], Scott N. Williamson[47], Sara Morris [48], Adam L. Atchley [49], Richard Essery[50], Benjamin R. K. Runkle [51], David Holl [16], Laura D. Riihimaki [35,52], Hiroki Iwata [53], Edward A. G. Schuur [54], Christopher J. Cox [48], Andrey A. Grachev[55], Joseph P. McFadden[56], Robert S. Fausto [33], Mathias Göckede [57], Masahito Ueyama [58], Norbert Pirk [59], Gijs de Boer [48,52,60], M. Syndonia Bret-Harte [7], Matti Leppäranta[61], Konrad Steffen[30,66], Thomas Friborg [40], Atsumu Ohmura[39], Colin W. Edgar[7], Johan Olofsson [62] & Scott D. Chambers[63]

[1]Department of Evolutionary Biology and Environmental Studies, University of Zurich, Winterthurerstrasse 190, 8057 Zurich, Switzerland. [2]Department of Biology, McGill University, 1205 Docteur Penfield, H3A 1B1 Montreal, QC, Canada. [3]Low-Carbon and Climate Impact Research Centre, School of Energy and Environment, City University of Hong Kong, Tat Chee Ave, Kowloon Tong, Hongkong, People's Republic of China. [4]Environmental Studies Program, Hamilton College, 198 College Hill Rd, Clinton, NY, USA. [5]Permafrost Research Section, Alfred-Wegener-Institute, Telegrafenberg 14473 Potsdam, Germany. [6]Département de géographie, Université de Montréal, 2900 Edouard Montpetit Blvd, Montreal, QC H3T 1J4, Canada. [7]Institute of Arctic Biology, University of Alaska Fairbanks, 2140 Koyukuk Dr, Fairbanks, AK, USA. [8]Institute of Terrestrial Ecosystems, ETH Zurich, CHN, Universitätstrasse 16, 8006 Zurich, Switzerland.

[9]Land Change Science Unit, Swiss Federal Institute for Forest, Snow and Landscape Research (WSL), Zürcherstrasse 111, 8903 Birmensdorf, ZH, Switzerland. [10]Istituto Nazionale di Geofisica e Vulcanologia, Via di Vigna Murata, 605 Rome, Italy. [11]Department of Geography, University of Colorado, Boulder, CO, USA. [12]School of Geographical Sciences, University of Bristol, University Rd, Bristol, UK. [13]Department for Sustainability, ENEA, Via Enrico Fermi 45, Frascati, Italy. [14]UK Centre for Ecology & Hydrology (UKCEH), MacLean Bldg, Benson Ln, Crowmarsh Gifford, Wallingford, UK. [15]Department of Hydrology and Water Management, Faculty of Earth Sciences and Spatial Management, Nicolaus Copernicus University, Lwowska 87-100 Toruń, Poland. [16]Center for Earth System Research and Sustainability (CEN), University of Hamburg, Bundesstrasse 53, 20146 Hamburg, Germany. [17]Finnish Meteorological Institute, Erik Palménin aukio 1, 00560 Helsinki, Finland. [18]Geography Department, Humboldt-Universität zu Berlin, Unter den Linden 6, 10117 Berlin, Germany. [19]University of Wisconsin-Madison, Madison, WI, USA. [20]School of Agriculture and Environment, The University of Western Australia, 35 Stirling Hwy, Crawley WA 6009 WA, Australia. [21]Department of Environment and Minerals, Greenland Institute of Natural Resources, Kivioq 2, Nuuk 3900, Greenland. [22]Department of Ecoscience, Aarhus University, Nordre Ringgade 1, 8000 Aarhus C, Denmark. [23]Environmental Science Division, Argonne National Laboratory, 9700 S Cass Ave, Lemont, IL, USA. [24]Department of Meteorology and Climatology, Faculty of Earth Sciences and Spatial Management, Nicolaus Copernicus University, Lwowska 87-100 Toruń, Poland. [25]Center for Biogeochemistry of the Anthropocene, Department of Geosciences, University of Oslo, Sem Sælands vei 1, 0371 Oslo, Norway. [26]Department of Physical Geography and Ecosystem Science, Lund University, Geocentrum II, Sölvegatan 12, 223 62 Lund, Sweden. [27]University of Nebraska - Lincoln, 1400 R St, Lincoln, NE, USA. [28]Oulanka Research Station, University of Oulu, Pentti Kaiteran katu 1, 90570 Oulu, Finland. [29]GFZ German Research Centre for Geosciences, Wissenschaftspark Albert Einstein, Telegrafenberg 14473 Potsdam, Germany. [30]Swiss Federal Institute for Forest, Snow, and Landscape Research (WSL), Zürcherstrasse 111, 8903 Birmensdorf, ZH, Switzerland. [31]Cold Regions Research Centre, Wilfrid Laurier University, 75 University Ave W, Waterloo, ON, Canada. [32]Harold Hamm School of Geology and Geological Engineering, University of North Dakota, Grand Forks, ND, USA. [33]Department of Glaciology and Climate, Geological Survey of Denmark and Greenland (GEUS), Øster Voldgade 10, 1350 Copenhagen, Denmark. [34]Jet Propulsion Laboratory, CalTech, 4800Oak Grove Dr, Pasadena, CA, USA. [35]NOAA Global Monitoring Laboratory, 325 Broadway, Boulder, CO, USA. [36]National Ecological Observatory Network, Battelle, 1685 38th St #100, Boulder, CO, USA. [37]Department of Atmospheric and Oceanic Sciences, University of Wisconsin-Madison, 1225 W Dayton St, Madison, WI, USA. [38]Alaska Biological Research, Inc, 2842Goldstream Rd, Fairbanks, AK, USA. [39]Institute for Atmospheric and Climate Science, ETH Zurich, CHN, Universitätstrasse 16, 8006 Zurich, Switzerland. [40]Department of Geosciences and Natural Resource Management, University of Copenhagen, Rolighedsvej 23, 1958 Frederiksberg, Denmark. [41]Department for Sustainability, ENEA, Lungotevere Grande Ammiraglio Thaon di Revel, 76, Rome, Italy. [42]Department of Chemistry, Université Laval, Pavillon Alexandre-Vachon, 1045 Av. de la Médecine, G1V 0A6 Québec, QC, Canada. [43]Takuvik Laboratory, CNRS-INSU, Département de Biologie, Université Laval, Pavillon Alexandre-Vachon, 1045 Av. de la Médecine, G1V 0A6 Québec, QC, Canada. [44]Copernicus Institute of Sustainable Development, Utrecht University, Vening Meinesz Building, Princetonlaan 8a, 3584 CB Utrecht, The Netherlands. [45]Centre for African Conservation Ecology, Nelson Mandela University, University Way, Summerstrand, Gqeberha, 6019 Port Elizabeth, South Africa. [46]Department of Biological Sciences, University of Notre Dame, 100 Galvin Life Sciences, Notre Dame, IN, USA. [47]Polar Knowledge Canada, Canadian High Arctic Research Station, 1 rue Uvajuq place, CP 2150 Cambridge Bay, NU, Canada. [48]NOAA Physical Sciences Laboratory, 325 Broadway, Boulder, CO, USA. [49]Los Alamos National Laboratory, Bikini Atoll Rd., SM 30, Los Alamos, NM, USA. [50]School of Geosciences, University of Edinburgh, Drummond St, Edinburgh EH8 9XP, UK. [51]Department of Biological & Agricultural Engineering, University of Arkansas, 1164 W Maple St, Fayetteville, AR, USA. [52]CIRES (Cooperative Institute for Research in Environmental Sciences), 216 UCB, University of Colorado Boulder Campus, Boulder, CO, USA. [53]Department of Environmental Science, Shinshu University, 3 Chome-1-1 Asahi, Matsumoto, Nagano 390-8621, Japan. [54]Center for Ecosystem Science and Society, Northern Arizona University, S San Francisco St, Flagstaff, AZ, USA. [55]DEVCOM Army Research Laboratory, Owen Rd, White Sands Missile Range, New Mexico, NM, USA. [56]Department of Geography and Earth Research Institute, University of California Santa Barbara, 5816Ellison Hall, Isla Vista, CA, USA. [57]Department of Biogeochemical Signals, Max Planck Institute for Biogeochemistry, Hans-Knöll-Straße 10, 07745 Jena, Germany. [58]Osaka Metropolitan University, Sakai, Kita Ward, Umeda, 1 Chome–2 – 2-600, Osaka, Japan. [59]Department of Geosciences, University of Oslo, Sem Sælands vei 1, 0371 Oslo, Norway. [60]IRISS (Integrated Remote and In Situ Sensing), University of Colorado, Boulder, CO, USA. [61]University of Helsinki, Yliopistonkatu 4, 00100 Helsinki, Finland. [62]Department of Ecology and Environmental Science, Umeå University, Linnaeus väg 4-6, 907 36 Umeå, Sweden. [63]ANSTO Lucas Heights, New Illawarra Rd, Lucas Heights NSW, 2234 Sydney, NSW, Australia. [64]These authors contributed equally: Jin-Soo Kim, Raleigh Grysko, Heather Kropp, Inge Grünberg, Vitalii Zemlianskii, Oliver Sonnentag, Eugénie S. Euskirchen, Merin Reji Chacko. [65]These authors jointly supervised this work: Jacqueline Oehri, Gabriela Schaepman-Strub. [66]Deceased: Konrad Steffen. ✉e-mail: jacqueline.oehri@gmail.com; gabriela.schaepman@ieu.uzh.ch

