## [Peer Review File · Nature Communications]

Vegetation Type is an Important Predictor of the Arctic Summer Land Surface Energy BudgetREVIEWER COMMENTS

Reviewer #1 (Remarks to the Author):

This manuscript evaluates the main factors that control the surface energy budget across the Arctic and finds that (a) vegetation type is a key predictor of the surface energy budget and (b) that vegetation type also exerts some control on the timing of the beginning of the summer-regime. The results rely on an admittedly relatively small set of tower site (small because there simply are not that many Tower Sites in the Arctic) data to complete their analysis.

The analysis is well done and the manuscript is put together well. The topic is of relevance to the broader earth system science community. The figures and tables are clear and appropriate. Overall, I found the manuscript enjoyable to read. My recommendation is that the paper be accepted pending the minor revisions outlined below. Most of my recommendations relate to clarity and to refining the main messages of the study.

Recommendations:

1. I don't think that the outcomes of this paper support the second part of this statement in the abstract. "Our results indicate complex shifts in surface energy fluxes with land-cover transitions and a lengthening summer season and contribute to a better representation of Arctic vegetation functions in Earth system models." That is, the results may inform the future direction of Earth System Models, but nothing in the paper actually contributes to a better representation of Arctic vegetation in these models.

2. One question that I had as I read through the manuscript was whether or not this result, namely that vegetation type is an important predictor of energy fluxes in the Arctic, is surprising or if it is expected. As I look through the list of drivers, I started to think that, despite what the authors state, maybe the result isn't that surprising. For example, if you consider vegetation type relative to some of the other predictors, I think you could anticipate a priori which ones would be important or not. For example, precipitation shows up as being relatively unimportant, which makes intuitive sense because the Arctic tends to be energy-limited rather than moisture-limited. Basically, there is almost always enough water around for plants to transpire freely. You might expect cloud cover to play a large role on Rnet, but it doesn't probably because it is cloudy a large fraction of the time across most of the Arctic. Anyway, it would probably be useful to include some discussion about when / where the results are as one would expect and when / where the results are not as expected. Even if it is mostly as expected, the quantification of the impact of different vegetation types is important.

3. Line 126: "The terrestrial Arctic is subject to extreme biogeophysical changes ...". Personally, I would avoid terms like 'extreme' in statements like this. How do you define extreme? Compared to what?

4. Lines 145-149: These statements are all correct. ESMs typically include only a few plant types for the Arctic region. Of course, ESMs only include a few plant types for all regions. I don't know that it would be possible, but one question these statements raise for me is whether or not the Arctic region is particularly deficient in its representation of plant type variety or whether the same problem would be found if one partitioned variance for regions outside of the Arctic. If there was an argument that implied that the Arctic was particularly problematic, that would be a powerful statement and would provide strong justification for increased work on Arctic plant types in ESMs.

5. Lines 232-245: I'm not sure I'm convinced by the argument in this paragraph that the difference among vegetation types is bigger than between glacier and vegetation types. Based on my reading of Table 2, this statement mainly comes from comparing barren tundra to glaciers. However, isn't the barren tundra almost exclusively in the far, far

northern Arctic (Canadian archipelago?), where there is very little incoming solar radiation and maybe not much precipitation (not sure about that). The very low LH in the barren tundra, isn't really a result of the vegetation type, I would guess, it is a result of the completely inhospitable climate for plants. So, characterizing the differences in surface energy fluxes between barren tundra and other veg types and glaciers as because of the vegetation type, as this paragraph implies, seems to me to not really be the right interpretation. The normalization to net radiation helps, but the extremely low incoming solar for barren tundra makes it hard to put it onto any sort of even footing with the other types.

6. Seasonality of Arctic SEB across veg types: I found the transition from the prior section to be really abrupt. The way it currently reads, it almost feels as if this is a new and independent study, with the first study relating to how vegetation types is a main predictor of summer SEB fluxes and this study being an assessment of the seasonality of the Arctic SEB. Other than the connection that both topics are within the realm of the SEB, it's not clear how these two topics are related. Some explanatory text that connects the two topics would I think help the reader. The connection between the two topics is clearer in the discussion section.

7. Related to (5) and I think partly contributing to the feeling of discontinuity is the following statement in the abstract: "Net radiation, sensible and ground heat fluxes show an unexpected early start of summer-regime (when daily mean values exceed 0 Wm^{-2}), preceding the end of snowmelt by 56, 33, and 39 days, respectively." First, is this really unexpected? I've never thought much about how to define the summer period, but it doesn't necessarily seem surprising to me that net radiation, sensible and ground heat fluxes exceed zero before all the snow has melted. But, more importantly, is this the real salient point of the seasonality analysis. I think the more salient point is that the timing of when these fluxes transition to being > 0 relative to the end of the snow melt season shows some strong dependence on vegetation type, implying that vegetation type is partly controlling how snow melt proceeds. This main point also ties in more strongly with the other main point that vegetation type is one of the main predictors of mean fluxes.

8. The paragraph starting on line 334 is good. It clearly explains how environmental conditions can affect vegetation type and also how vegetation type can affect environmental conditions. I would agree with the perspective that it is both reasons that explain the 'predictive power' of vegetation type. However, I think the summary statement in that paragraph is worded incorrectly. I think the results of this study indicate that climate models could potentially be improved with an expanded and more comprehensive treatment of Arctic vegetation types and/or a more explicit representation of how environmental conditions can help dictate vegetation type (e.g., in the context of dynamic vegetation or demographic models). This isn't the same thing as saying that models can be improved by incorporating vegetation type as a predictor. Vegetation type in and of itself does not determine the simulated SEB, it is the characteristics of each vegetation type (height, LAI, leaf shape, leaf physiology, leaf reflectance and transmittance, phenological timing, etc) that will help determine the SEB. The implicit statement here is that the existing representation of Arctic vegetation type diversity is insufficient to accurately capture the range of vegetation impacts on SEB that are observed.

9. Line 408: I would say 'additional' rather than 'different'.

Reviewer #2 (Remarks to the Author):

This is a well written, well-conceived, comprehensive study that provides useful information on the link between Arctic vegetation types and energy balance. To my knowledge, this is the first report of the link between a single vegetation classification and the respective energy balances for the pan Arctic region.

The authors are careful in their analysis and in their interpretation of results and they recognize and acknowledge that there is a circular relationship between environment and vegetation type that bears on energy balance. Vegetation is controlled by environmental factors that affect energy balance (e.g., soil moisture and water table) and vegetation also affects environmental factors that affect energy balance (e.g., snow accumulation). As a result, there are many situations where these results will help understand current and predict future energy balance.

The vegetation classifications are very broad and miss finer division of vegetation types and categories that can have profound effect on the local energy balance. Interestingly, these more specific vegetation types are often the vegetation associated with the energy balance data used. The general vegetation types used are often comprised of a number of land surface and vegetation types that vary widely in their energy balance characteristics.

Manuscript: NCOMMS-22-08311A

Authors: Oehri J. et al.

Title: Vegetation Type is an Important Predictor of the Arctic Summer Land Surface Energy Budget

Point-by-point response to reviewer comments

We thank both reviewers for their constructive feedback that helped to improve the quality and clarity of our manuscript. We address the reviewers' comments point-by-point in blue color below.

Reviewer #1 (Remarks to the Author):

This manuscript evaluates the main factors that control the surface energy budget across the Arctic and finds that (a) vegetation type is a key predictor of the surface energy budget and (b) that vegetation type also exerts some control on the timing of the beginning of the summer-regime. The results rely on an admittedly relatively small set of tower site (small because there simply are not that many Tower Sites in the Arctic) data to complete their analysis.

The analysis is well done and the manuscript is put together well. The topic is of relevance to the broader earth system science community. The figures and tables are clear and appropriate. Overall, I found the manuscript enjoyable to read. My recommendation is that the paper be accepted pending the minor revisions outlined below. Most of my recommendations relate to clarity and to refining the main messages of the study.

Thank you for this positive feedback. We appreciate that our manuscript was well received, and that the relevance of our study was recognized. We address your thoughtful comments below and in the revised manuscript.

Recommendations:

1. I don't think that the outcomes of this paper support the second part of this statement in the abstract. "Our results indicate complex shifts in surface energy fluxes with land-cover transitions and a lengthening summer season and contribute to a better representation of Arctic vegetation functions in Earth system models." That is, the results may inform the future direction of Earth System Models, but nothing in the paper actually contributes to a better representation of Arctic vegetation in these models.

Thank you, we agree with this comment and rephrased the sentence accordingly (lines 123-126):

"Our results indicate complex shifts in surface energy fluxes with land-cover transitions and a lengthening summer season, and highlight the potential for improving future Earth system models via a refined representation of Arctic vegetation types."

2. One question that I had as I read through the manuscript was whether or not this result, namely that vegetation type is an important predictor of energy fluxes in the Arctic, is surprising or if it is expected. As I look through the list of drivers, I started to think that, despite what the authors state, maybe the result isn't that surprising. For example, if you consider vegetation type relative to some of the other predictors, I think you could anticipate a priori which ones would be important or not. For example, precipitation shows up as being relatively unimportant, which makes intuitive sense because the Arctic tends to be energy-limited rather than moisture-limited. Basically, there is almost always enough water around for plants to transpire freely. You might expect cloud cover to play a large role on Rnet, but it doesn't probably because it is cloudy a large fraction of the time across most of the Arctic. Anyway, it would probably be useful to include some discussion about when / where the results are as one would expect and when / where the results are not as expected. Even if it is mostly as expected, the quantification of the impact of different vegetation types is important.

We appreciate this thoughtful comment.

Indeed, we agree that it is to be expected that vegetation type plays an important role for surface energy fluxes, in particular in summer, when local control by the land surface is relatively strong (cf. main text lines 153-156 and lines 174-176). This is not least because vegetation types differ in SEB-relevant traits and because vegetation types are correlated with distinct environmental conditions (lines 347-353). However, prior to our study it was not clear how important vegetation type is compared with other environmental drivers at the seasonal and circumarctic scale: Discussions among the coauthors of this study, with backgrounds ranging from ecology to geography and climate sciences, showed that some were surprised by the relatively large predictive power of vegetation type in comparison with other drivers, in particular cloud cover and latitude (cf. Figure 1).

The finding that cloud cover does not explain much variation is counter-intuitive, but indeed likely due to the reason you pointed out: even though clouds have a large effect on surface radiative fluxes (Lund et al. 2017), the spatial variability of that effect is limited by the persistence of high fractional cloud cover across large parts of the Arctic, especially in summer and autumn (Shupe et al. 2011). Hence, one would see the cloud cover effect in the overall forcing but not as a primary cause of variability among study sites. As you pointed out, the relatively small predictive power of precipitation for most SEB fluxes (cf. Supplementary Figure 5), compared to temperature and related predictors (especially summer warmth and continentality), could indicate that our sites are energy-limited rather than water-limited (McVicar et al. 2012), or it could be that the precipitation regime is already accounted for by our vegetation types. Another expected result was that summer net radiation is largely explained by snow cover duration and latitude (Cox et al. 2017) whereas for sensible, latent and ground heat fluxes, which more directly depend on an interplay between vegetation functions (including evapotranspiration) and soil characteristics, these vegetation and soil-related variables are also more important (Eugster et al. 2000).

We added this more nuanced discussion of expected and unexpected patterns in our results to the main text (lines 362-369):

“Our variance partitioning analysis also reveals a low predictive power of cloud cover, which is counter-intuitive^{15,38}, but can be explained by the persistence of high fractional cloud cover and its consequent low spatial variability across large parts of the Arctic in summer⁴³. Therefore, even though cloud cover has a large effect on the overall surface radiative forcing, it does not explain much variability among our study sites^{15,38,43}. The relatively small predictive power of precipitation could be indicative of energy-limited, rather than water-limited systems⁴⁴

or be due to the fact that the precipitation regime is already captured by other variables in our analysis, including vegetation type¹¹.”

3. Line 126: “The terrestrial Arctic is subject to extreme biogeophysical changes ...”. Personally, I would avoid terms like ‘extreme’ in statements like this. How do you define extreme? Compared to what?

Thank you for this comment, indeed, ‘extreme’ is not a very precise description. We wanted to express that the trends and variability in surface air temperature in the Arctic are about two to four times larger than the global average (Serreze & Barry 2011, Box et al. 2019, Rantanen et al. 2021, Chylek et al. 2022), resulting in the change into an ‘unprecedented’ state of the Arctic biophysical system (Box et al. 2019, McCrystall et al. 2021).

We now have changed the language on lines 129-130 to describe more precisely what we want to communicate:

“As the Arctic warms at a fast pace above the global average^{1,2}, changes to a broad range of biogeophysical variables are being reported and projected^{3,4}.”

4. Lines 145-149: These statements are all correct. ESMs typically include only a few plant types for the Arctic region. Of course, ESMs only include a few plant types for all regions. I don’t know that it would be possible, but one question these statements raise for me is whether or not the Arctic region is particularly deficient in its representation of plant type variety or whether the same problem would be found if one partitioned variance for regions outside of the Arctic. If there was an argument that implied that the Arctic was particularly problematic, that would be a powerful statement and would provide strong justification for increased work on Arctic plant types in ESMs.

This is an excellent point. Indeed, Earth system models include only a few plant functional types for all regions on Earth, not only the Arctic (Dallmeyer et al. 2019).

Nevertheless, while additional, distinct functional types for non-Arctic regions are being implemented, Arctic vegetation continues to be largely represented by one or two types only (Sulman et al. 2021). For example, the Community Land Model newest version 5 specifies 15 main plant functional types and additionally 8 ‘crop functional types’ (that can be irrigated or not, resulting in 16 CFT), whereas Arctic vegetation is represented by one type, the ‘C3 arctic grass’ (Lawrence et al. 2019).

Since terrestrial vegetation is critical for Earth system dynamics and among the largest sources of uncertainty in climate change predictions (Wullschleger et al. 2014), we think that ESM projections would generally benefit from a more comprehensive and detailed representation of vegetation in all regions on Earth, not only the Arctic.

However, given that

1. in the Arctic, the uncertainty in SEB and climate projections is especially high (Duncan et al. 2020, Block et al. 2020),
2. the Arctic is especially sensitive to climate change, i.e. changing at a pace above the global average (Box et al. 2019, Wullschleger et al. 2014),
3. the Arctic system exerts key land-climate feedbacks important for global climate dynamics, including the permafrost-climate-carbon feedback (Lawrence et al. 2019, Sulman et al. 2021),

4. the Arctic is across large areas dominated by vegetation with unique properties relevant for land-atmosphere feedbacks (including moss and lichen, Wullschleger et al. 2014, Sulman et al. 2021),

we think the Arctic is one of the priority areas to further develop the ESM land surface schemes, among others by a more comprehensive representation of vegetation types and their effects on the climate system.

We added this argument to the Discussion section (lines 432-437):

“The Arctic system is highly sensitive to climate change^{3,18}, exerts key land-feedbacks relevant for global climate dynamics^{2,54} and harbors a range of vegetation types with unique traits including mosses and lichens¹⁸⁻²⁰. For the future, a widespread redistribution of Arctic vegetation is predicted^{6,55,56}. Understanding and predicting how these changes in turn affect climate is essential for reducing persisting uncertainties in climate projections^{2,16,18}.”

5. Lines 232-245: I'm not sure I'm convinced by the argument in this paragraph that the difference among vegetation types is bigger than between glacier and vegetation types. Based on my reading of Table 2, this statement mainly comes from comparing barren tundra to glaciers. However, isn't the barren tundra almost exclusively in the far, far northern Arctic (Canadian archipelago?), where there is very little incoming solar radiation and maybe not much precipitation (not sure about that). The very low LH in the barren tundra, isn't really a result of the vegetation type, I would guess, it is a result of the completely inhospitable climate for plants. So, characterizing the differences in surface energy fluxes between barren tundra and other veg types and glaciers as because of the vegetation type, as this paragraph implies, seems to me to not really be the right interpretation. The normalization to net radiation helps, but the extremely low incoming solar for barren tundra makes it hard to put it onto any sort of even footing with the other types.

Thank you, this is a very important point.

It is true that the strongest differences in surface energy fluxes were found between barren tundra and other vegetation types, especially in the case of sensible heat flux (Supplementary Table 4). However, the barren tundra in our study is located on 67.13° North (see “Kan_B” site in Supplementary Table 2) and is therefore a rather southern example of this vegetation type (in fact, it is more south than two of the boreal peat bog sites). One advantage of our dataset is that vegetation types and glacier sites are not strongly confounded with latitude, i.e. vegetation and glacier site locations are well mixed along a gradient from North to South (see Supplementary Table 2). Hence, we do think that the differences we found are likely to be at least partly due to the barren vegetation type rather than the geographical position. We think it is important to highlight the potentially strong differences between barren types and other vegetation types, not least because this type currently covers a large fraction (about one fifth) of the terrestrial Arctic and therefore we encourage increased sampling in these regions (cf. Supplementary Analysis). However, it is indeed appropriate to highlight that our results must be interpreted with caution since we have only one site covered with this vegetation type.

We have changed the text accordingly on lines 396-400:

“However, barren tundra is only covered by one study site in our in situ observations dataset and generally underrepresented in the surface energy budget literature (Supplementary Analysis). Therefore, our results need to be interpreted with caution and further research is needed to assess potential climate feedbacks with more certainty.”

Importantly, in the case of latent heat flux, it is not barren tundra but boreal peat bog that shows strongest differences with a variety of other vegetation types (including barren tundra, Supplementary Table 4).

We think there are two important take home messages from our pairwise difference analysis:

1. Most significant differences among vegetation types, as well as between vegetation and glacier sites are found for latent and sensible heat fluxes, whereas net radiation and ground heat flux seem to be more similar - even between glacier and vegetation types.
2. Whereas it is obvious and expected that glacier and vegetation sites show significant differences in SEB fluxes (e.g. between 38.5 and 66.8 Wm^{-2} for latent heat flux), we demonstrate that differences among vegetation types can be in the same range (e.g. between 28.3 and 74.9 Wm^{-2} for latent heat flux, Supplementary Table 4).

Hence, our results suggest that for modeling summer SEB fluxes, especially for H and LE, differentiating between vegetation types is as important as differentiating between glacier and vegetated land cover. Given our small sample size and therefore low statistical power, these results are rather conservative.

We have changed the text on lines 373-378 to explain our points in a more deliberate way:

“Interestingly, surface energy flux differences among vegetation types can be as high or even higher than among glacier and vegetation sites, especially in the case of sensible and latent heat fluxes (Supplementary Table 4).

For example, in the case of latent heat flux, significant absolute differences between glacier and vegetation types are in a similar range (38.5-66.8 Wm^{-2}) as differences among vegetation types only (range: 28.3-74.9 Wm^{-2} ; Supplementary Table 4).”

6. Seasonality of Arctic SEB across veg types: I found the transition from the prior section to be really abrupt. The way it currently reads, it almost feels as if this is a new and independent study, with the first study relating to how vegetation types is a main predictor of summer SEB fluxes and this study being an assessment of the seasonality of the Arctic SEB. Other than the connection that both topics are within the realm of the SEB, it's not clear how these two topics are related. Some explanatory text that connects the two topics would I think help the reader. The connection between the two topics is clearer in the discussion section.

Thank you for this comment. Indeed, the transition was quite abrupt. We changed the first paragraph accordingly to highlight the connection of our magnitude and seasonality analyses across vegetation types. Specifically, we highlight our focus on seasonality to assess the role of vegetation type for the cumulative surface energy budget (lines 276-281):

“The cumulative summer energy budget depends on both the magnitude and the seasonality of SEB fluxes. In the Arctic, the seasonality of SEB fluxes is largely governed by the seasonal changes in incoming radiation, temperature, cloud cover and associated melt and onset of snow cover^{15,38}. To assess the role of vegetation type for the seasonal change in SEB-flux magnitudes, we quantify the timing of SEB-flux ‘summer-regimes’ relative to the timing of the snow-free date in spring and the snow-onset date in autumn^{39,40}.”

7. Related to (5) and I think partly contributing to the feeling of discontinuity is the following statement in the abstract: “Net radiation, sensible and ground heat fluxes show an unexpected early start of summer-regime (when daily mean values exceed 0 Wm^{-2}), preceding the end of snowmelt by 56, 33, and 39 days, respectively.” First, is this really unexpected? I’ve never thought much about how to define the summer period, but it doesn’t necessarily seem surprising to me that net radiation, sensible and ground heat fluxes exceed zero before all the snow has melted. But, more importantly, is this the real salient point of the seasonality analysis. I think the more salient point is that the timing of when these fluxes transition to being > 0 relative to the end of the snow melt season shows some strong dependence on vegetation type, implying that vegetation type is partly controlling how snow melt proceeds. This main point also ties in more strongly with the other main point that vegetation type is one of the main predictors of mean fluxes.

Thank you for this comment, it points out an important message we should highlight more in our manuscript.

We agree that it is not totally unexpected that SEB fluxes turn from negative to positive before all snow has melted in spring. Nevertheless, previous research highlights the importance of the timing of snowmelt and onset for the SEB and vegetation activity in the Arctic (references on lines 405-406). However, there are also potentially important effects of vegetation types on the seasonality of SEB fluxes and timing of snowmelt and onset (cf. lines 407-409).

We therefore think our seasonality analysis has two important take home messages:

1. The start of season occurs substantially earlier than snowmelt for most SEB fluxes, while the SEB-flux end-of-summer timings are more aligned with snow onset (except for ground heat flux). Thus, for the start of season timing, additional factors are important, whereas end of season timings appear to be more directly controlled by the timing of snow cover and associated atmospheric drivers.
2. Start of season timing for Rnet and H, and end of season timing of G are highly variable among vegetation types, which supports the notion that vegetation types differentially affect the trapping and accumulation of snow, and the relative timing of SEB-flux summer regime transitions. A recent study highlights that vegetation types can even affect the SEB-flux directions when still snow covered (Domine et al. 2021).

We agree that the more salient point about our analysis is the second point. We therefore rewrote the abstract to highlight this more clearly (lines 120-123):

“The timing of SEB-flux summer-regimes (when daily mean values exceed 0 Wm^{-2}) relative to snow-free and -onset dates varies substantially depending on vegetation type, implying vegetation controls on snow-cover and SEB-flux seasonality.”

8. The paragraph starting on line 334 is good. It clearly explains how environmental conditions can affect vegetation type and also how vegetation type can affect environmental conditions. I would agree with the perspective that it is both reasons that explain the ‘predictive power’ of vegetation type. However, I think the summary statement in that paragraph is worded incorrectly. I think the results of this study indicate that climate models could potentially be improved with an expanded and more comprehensive treatment of Arctic vegetation types and/or a more explicit representation of how environmental conditions can help dictate vegetation type (e.g., in the context of dynamic vegetation or demographic models). This isn’t the same thing as saying that models can be improved by incorporating vegetation type as a predictor. Vegetation type in and of itself does not determine the simulated SEB, it is the

characteristics of each vegetation type (height, LAI, leaf shape, leaf physiology, leaf reflectance and transmittance, phenological timing, etc) that will help determine the SEB. The implicit statement here is that the existing representation of Arctic vegetation type diversity is insufficient to accurately capture the range of vegetation impacts on SEB that are observed. Thank you, yes, your wording is a much more precise description of what we wanted to communicate here. We therefore changed the main text accordingly on lines 358-361:

“Nevertheless, our results highlight the potential for improving the predictions of Arctic surface energy fluxes, specifically of summer latent and sensible heat fluxes, by a more comprehensive treatment of vegetation types and how environmental conditions interact with associated vegetation functional traits^{18,19}.”

9. Line 408: I would say ‘additional’ rather than ‘different’.

Thank you, this is a good point. We changed the terminology on lines 441-445:

“Here, we provide quantitative evidence of the importance of vegetation types for predicting Arctic surface energy budgets at circumpolar scale and support recent calls for refined accounting of high-latitude vegetation types and associated vegetation functions in land surface components of Earth system models^{18,19,58}.”

Reviewer #2 (Remarks to the Author):

This is a well written, well-conceived, comprehensive study that provides useful information on the link between Arctic vegetation types and energy balance. To my knowledge, this is the first report of the link between a single vegetation classification and the respective energy balances for the pan Arctic region.

Thank you for this positive assessment, we appreciate that the novelty of our study was recognized.

The authors are careful in their analysis and in their interpretation of results and they recognize and acknowledge that there is a circular relationship between environment and vegetation type that bears on energy balance. Vegetation is controlled by environmental factors that affect energy balance (e.g., soil moisture and water table) and vegetation also affects environmental factors that affect energy balance (e.g., snow accumulation). As a result, there are many situations where these results will help understand current and predict future energy balance. Thank you for this comment, we agree and indeed hope to contribute to a better understanding of these complex relationships between environment, vegetation, and climate.

The vegetation classifications are very broad and miss finer division of vegetation types and categories that can have profound effect on the local energy balance. Interestingly, these more specific vegetation types are often the vegetation associated with the energy balance data

used. The general vegetation types used are often comprised of a number of land surface and vegetation types that vary widely in their energy balance characteristics.

Thank you for this comment. We agree that the vegetation categories we used are broad and often comprise several vegetation types that vary in their energy balance characteristics (see Supplementary Analysis).

We could not use a more detailed classification of vegetation types because of the low number of tower sites available in the Arctic. Nevertheless, we think the physiognomic CAVM classes are meaningful because they differ in key-vegetation traits that have been shown to be important for SEB fluxes, including vegetation height, albedo, and productivity (cf. main text lines 350-353). Importantly, we derived the 'Vegetation type' variable for each study site from *in situ* site descriptions in the literature (Supplementary Tables 2 and 6), following the CAVM class description. Figure 1 shows that indeed our literature-based, and therefore, more local vegetation type ('Vegetation type') is a more powerful predictor of SEB fluxes, compared to the vegetation type predicted by and extracted from the CAVM raster map ('CAVM type'), which represents the dominant vegetation at the landscape-scale (500m radius around study site).

The point you raise is of general importance and has been highlighted in previous studies (Wullschleger 2014). We think this point deserves more attention and therefore we added a statement in the main text (lines 367-371):

“Further research will be needed to identify the specific vegetation trait combinations that are most relevant for the SEB in the Arctic, including traits not represented by the main CAVM classes we used (Supplementary Analysis), and to better constrain the implications of changing Arctic vegetation productivity, plant height and shrub abundance for the SEB⁶⁻⁸.“

References

1. Block, K. et al. Climate models disagree on the sign of total radiative feedback in the Arctic. *Tellus A: Dynamic Meteorology and Oceanography*, 72(1), 1-14 (2020).
2. Box, J. E. et al. Key indicators of Arctic climate change: 1971–2017. *Environ. Res. Lett.* 14, 045010 (2019).
3. Chylek, P., Folland, C., Klett, J. D., Wang, M., Hengartner, N., Lesins, G., & Dubey, M. K. Annual Mean Arctic Amplification 1970- 2020: Observed and simulated by CMIP6 climate models. *Geophysical Research Letters*, e2022GL099371 (2022).
4. Cox, C. J. et al. Drivers and environmental responses to the changing annual snow 752 cycle of northern Alaska. *Bull. Amer. Meteor. Soc.* 98, 2559–2577 (2017).
5. Dallmeyer, A., Claussen, M., & Brovkin, V.. Harmonising plant functional type distributions for evaluating Earth system models. *Clim. Past*, 15(1), 335-366 (2019).
6. Domine, F., Fourteau, K., Picard, G., Lackner, G., Sarrazin, D., & Poirier, M. Thermal Bridging Through Branches of Snow-Covered Shrubs Cool Down Permafrost in Winter. Under Review, <https://doi.org/10.21203/rs.3.rs-679013/v1> (2021).
7. Duncan, B. N. et al. Space-based observations for understanding changes in the 696 arctic-boreal zone. *Rev. Geophys.* 58, (2020).
8. Eugster, W. et al. Land-atmosphere energy exchange in Arctic tundra and boreal forest: 686 available data and feedbacks to climate. *Glob. Chang. Biol.* 6, 84–115 (2000).
9. Lawrence, D. M. et al. The Community Land Model version 5: Description of new features, benchmarking, and impact of forcing uncertainty. *J. Adv. Model. Earth Syst.*, 11(12), 4245-4287 (2019).
10. Lund, M. et al. Spatiotemporal variability in surface energy balance across tundra, snow and ice in Greenland. *Ambio* 46, 81–93 (2017).
11. McCrystall, M. R., Stroeve, J., Serreze, M., Forbes, B. C. & Screen, J. A. New climate models reveal faster and larger increases in Arctic precipitation than previously projected. *Nat. Commun.* 12, 6765 (2021).
12. McVicar, T. R. et al. Global review and synthesis of trends in observed terrestrial near-surface wind speeds: Implications for evaporation. *J. Hydrol.*, 416, 182-205 (2012).
13. Myers-Smith, I. H. and Hik, D. S.: Shrub canopies influence soil temperatures but not nutrient dynamics: An experimental test of tundra snow-shrub interactions, *Ecol. Evol.*, 3, 3683–3700, <https://doi.org/10.1002/ece3.710> (2013).
14. Rantanen, M. et al. The Arctic has warmed four times faster than the globe since 1980, 12 July 2021, PREPRINT (Version 1) available at Research Square [<https://doi.org/10.21203/rs.3.rs-654081/v1>]
15. Serreze, M. C. & Barry, R. G. Processes and impacts of Arctic amplification: A research synthesis. *Glob. Planet. Change* 77, 85–96 (2011).
16. Shupe, M. D., et al. Clouds at Arctic atmospheric observatories. Part I: Occurrence and macrophysical properties. *J. Appl. Meteorol. Climatol.*, 50(3), 626-644 (2011).
17. Sulman, B. N. et al. Integrating Arctic Plant Functional Types in a Land Surface Model 704 Using Above- and Belowground Field Observations. *J. Adv. Model. Earth Syst.* 13, 705 (2021).
18. Wullschleger, S. D. et al. Plant functional types in Earth system models: past experiences and future directions for application of dynamic vegetation models in high-latitude ecosystems. *Ann. Bot.* 114, 1–16 (2014).

REVIEWERS' COMMENTS

Reviewer #1 (Remarks to the Author):

The authors have adequately addressed all my comments from the previous review and I have no additional comments.